# Enforcing Instruction Hierarchy via Augmented Intermediate Representations

## Abstract

Indirect prompt injection attacks are a critical security vulnerability in large language models (LLMs), allowing attackers to hijack model behavior by injecting malicious instructions within the input context. Recent defense mechanisms have leveraged an *Instruction Hierarchy* (IH) Signal – often implemented through special delimiter tokens or additive embeddings – to denote the privilege level of input tokens. However, these prior works typically inject the IH signal exclusively at the initial input layer, which we hypothesize limits its ability to effectively distinguish the privilege levels of tokens as it propagates through the different layers of the model. To overcome this limitation, we introduce a novel approach that injects the IH signal into the intermediate token representations within the network. Our method augments these representations with layer-specific trainable embeddings that encode the privilege information. Our evaluations across multiple models and training methods reveal that our proposal yields between $1.6\times$ and $9.2\times$ reduction in attack success rate on gradient-based prompt injection attacks compared to state-of-the-art methods, without significantly degrading the model's utility.

## 1 Introduction

Transformer (Vaswani et al., 2017) based large language models (LLMs) exhibit a notable sensitivity to specific tokens within their input context, allowing even a small subset to significantly influence the distribution of generated responses. While this characteristic underpins the flexibility of LLMs, it also introduces a critical vulnerability: *indirect prompt injection attacks* (Greshake et al., 2023). These attacks involve the strategic insertion of adversarial tokens into the LLM's context to override the user's intended instructions and compel the model to adhere to the adversary's commands instead. Recent research demonstrated the potential for such attacks to generate inaccurate information, lure users to harmful websites, and facilitate the exfiltration of sensitive data, including passwords and personal details (Greshake et al., 2023). This susceptibility poses a particularly significant challenge for agentic AI systems (Debenedetti et al., 2024), where LLMs are entrusted with executing complex tasks involving potentially untrusted data sources and websites, often without human oversight.

Several recent studies (Wallace et al., 2024; Chen et al., 2024a; Wu et al., 2024; Chen et al., 2024b) have proposed defense mechanisms aimed at making the model more robust to these prompt injection attacks. A key commonality among these approaches is the concept of an *instruction hierarchy* (IH). Rather than treating all input tokens uniformly, an IH framework assigns varying levels of importance or privilege to different tokens within the context. These privilege levels can then be leveraged to dictate the appropriate behavior when conflicting instructions arise. Prior works have explored different techniques for (a) injecting IH signals into the LLM and (b) training the LLM to recognize and respect these signals. This research focuses on enhancing the method of injecting the IH signal to the LLM. We observe that existing approaches primarily inject the IH signal *solely at the input level*, either by introducing novel delimiter tokens or by modifying the input token embeddings to encode IH information. We hypothesize that limiting the injection of this crucial information to the input layer constrains the signal's overall efficacy.

To address this limitation, we introduce Augmented Intermediate Representations (AIR). AIR distinguishes itself by injecting IH signals recurrently across all layers of the LLM, rather than confining it to the initial input layer. We posit that the consistent availability of IH signals at each processing stage

Figure 1: Illustration of prompt injection attack. By injecting malicious tokens $D'$ into the context window, an adversary can control the LLM's behavior, making it follow malicious instructions ($I'$) instead of the user's original instructions ($I$). $\mathcal{A}$ denotes the alignment function.

can facilitate a stronger enforcement of the intended instruction hierarchy and enable the training of models that are more robust to prompt injection attacks.

**Contributions.** The primary contributions of this work are outlined below:

1. We identify a critical limitation in existing prompt injection defense mechanisms: their reliance on injecting instruction hierarchy (IH) signals solely at the input level, which consequently restricts their overall effectiveness.
2. To address this limitation, we introduce Augmented Intermediate Representations (AIR). Our core insight is to inject IH signals recurrently across all layers of the LLM, thereby enabling a more robust enforcement of the intended instruction hierarchy.
3. Our empirical evaluations across multiple models, training setups, and evaluation datasets reveal that AIR consistently improves robustness, yielding a $1.6\times$ to $9.2\times$ reduction in ASR compared to previous methods on gradient based attacks, while only minimally impacting the model's utility.

## 2 PRELIMINARIES

To formally discuss the dynamics of indirect prompt injection attacks and defenses, we first establish a clear framework. This section defines the core components of our threat model, including the user, LLM, and the attacker, along with their respective objectives and interactions.

**Setup.** Our setup considers a benign user employing a large language model $\mathcal{M}$ to execute a task. This task is accomplished through the LLM's processing of user-provided instruction tokens $I$ and data tokens $\hat{D}$ that may originate from potentially untrusted sources, such as external websites or emails. We denote the LLM's resulting output as $O = \mathcal{M}(I + \hat{D})$. We further assume that the data tokens consist of benign tokens $D$ and adversarial tokens $D'$ controlled by an attacker i.e. $\hat{D} = D + D'$. To quantify how well the output follows the input, we define an alignment function $\mathcal{A}(O, I) \in [0, 1]$. Here, 0 indicates that $O$ does not follow $I$ and 1 signifies perfect alignment.

**Attacker's Goal.** The attacker's objective is to utilize the adversarial tokens $D'$ to manipulate the LLM's output such that it aligns with the attacker's instruction $I'$ instead of the user's instruction $I$. The attacker's goal can be formally expressed as maximizing $\mathcal{A}(O, I')$ by strategically selecting and injecting adversarial tokens $D'$ into the LLM's context window. For simplicity, we represent the sequence of adversarial tokens $D'$ as a combination of an adversarial prefix $D'_p$ and the adversarial instruction $I'$ i.e. $D' = D'_p + I'$.

**Illustrative Example.** Figure 1 shows an example of a successful prompt injection attack in the context of email summarization. The user's initial instruction ($I$) is to summarize unread emails. Benign data ($D$) might include legitimate emails, such as Email #1. However, an adversary can inject malicious tokens $D'$ by sending a crafted email (Email #2) containing an adversarial instruction $I'$ along with a suitable prefix $D'_p$. When the LLM processes this combined context, the injected adversarial instruction overrides the user's intent, leading the LLM to produce the output $O$: "You have no new emails.", breaking the alignment with the user's instructions ($I$) and making it follow the adversary's instruction ($I'$) instead.

**Defender's Goal.** The defender has two objectives. First, the defender aims to ensure that the LLM's response remains aligned with the user's intended instructions, even in the presence of malicious tokens, which can be expressed as maximizing $\mathcal{A}(O, I)$. Second, the defender seeks to maintain a high quality of the model's response in benign settings (i.e., even in the absence of an attack),

which can be denoted as maximizing a quality metric $\mathcal{Q}(O|I, D)$. In this context, the defender is typically the model provider. Thus, the defender's action space includes choices regarding the model's architecture (e.g., layer design, attention mechanisms) and the training process (e.g., data curation, training objectives).

# 3 RELATED WORK

The prompt injection attack was initially conceptualized in scenarios where an adversarial user, possessing the ability to directly prompt the LLM, attempts to override the intended system instructions (Perez & Ribeiro, 2022). This attack vector is referred to as *direct prompt injection*. Subsequently, a more covert variant, known as *indirect prompt injection*, was developed (Greshake et al., 2023). In this case, the attacker lacks the capability to directly interact with the LLM. Instead, they embed the attack within an external data source (e.g., documents, emails, or webpages) that the LLM ingests to generate responses to user prompts. While we primarily consider indirect prompt injection attacks in our paper, the insights behind our defense can be extended to direct prompt injection attacks as well. We proceed to discuss the various methodologies employed for generating prompt injection attacks, as well as prior research dedicated to defending against such attacks. Additional related work can be found in Appendix D.

## 3.1 ATTACKS

As outlined in Section 2, the attacker's primary objective is to identify an adversarial prefix $D'_p$ that compels the LLM's output to align with the attacker's intended instructions $I'$. Previous research has detailed several methods for constructing such adversarial prefixes. These methods can be broadly categorized into static attacks and optimization-based attacks.

**Static Attacks.** Static attacks rely on handcrafted prefixes that have been empirically demonstrated to deceive LLMs, causing them to prioritize the adversary's instructions over the user's. The *Ignore attack* (Perez & Ribeiro, 2022) exemplifies this approach by injecting phrases such as "Ignore previous instructions" (Fig 1). Completion attacks, on the other hand, insert a fabricated completion within the prefix, creating the illusion that the original query has already been addressed, thereby prompting the LLM to respond to the adversary's subsequent instructions. The escape separation attack involves inserting a sequence of escaped characters, such as "\n" and "\t", as the prefix.

**Gradient-based Attacks.** These attacks employ gradient-based optimization techniques to identify prefixes that maximize the likelihood of the LLM generating the adversary's desired response. Greedy Coordinate Gradient (GCG) (Zou et al., 2023) is a prominent example, where the attacker initializes the adversarial prefix $D'_p$ with a randomly selected set of tokens. A loss function $\mathcal{L}(D'_p)$ is then defined based on the output probability of the desired response: $\mathcal{L}(D'_p) = -\log p(O|I + D + D'_p + I')$. By iteratively optimizing $D'_p$ to minimize $\mathcal{L}(D'_p)$, GCG can identify a prefix that significantly increases the probability of the attacker's desired outcome. Several subsequent works have aimed to enhance the effectiveness of GCG. For instance, Zhang & Wei (2025) propose the use of momentum to improve GCG's performance. NeuralExec (Pasquini et al., 2024) employs a similar gradient-based optimization approach to execute prompt injection attacks. Unlike GCG, NeuralExec's adversarial prompt comprises both a prefix ($D'_p$) and a suffix ($D'_s$), i.e., $D' = D'_p + I' + D'_s$, which are both optimized using gradients. Astra (Pandya et al., 2025) optimizes the adversarial prefix to focus the model's attention on the attacker's instructions and uses this as a warm-start for GCG.

## 3.2 DEFENSES

A fundamental challenge identified in prior work is that LLMs often lack the ability to distinguish tokens originating from different sources, treating them with equal priority. This absence of privilege levels allows adversarial instructions to sometimes override legitimate user instructions, thereby facilitating prompt injection attacks. To address this issue, recent studies (Chen et al., 2024a; Wallace et al., 2024) propose structuring input tokens to assign varying levels of privilege to tokens from different sources (e.g., system, user, data). This privilege information can then be leveraged by the model to determine the appropriate response in scenarios involving conflicting instructions. Several defense mechanisms have been developed based on this core principle.

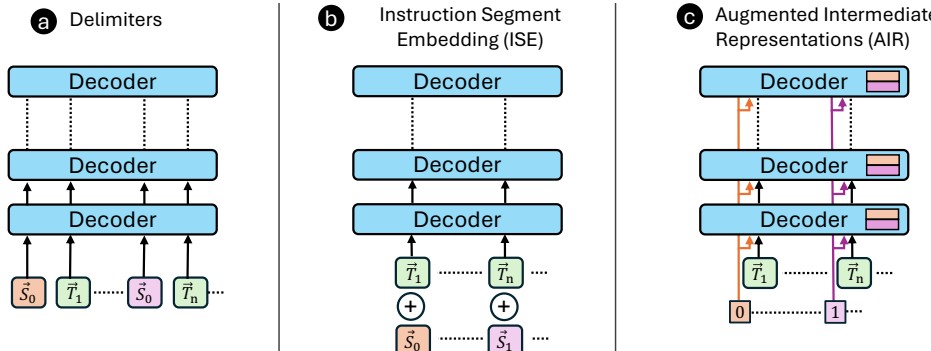

Figure 2: A comparison of different mechanisms for injecting Instruction Hierarchy (IH) signals into LLMs. Existing techniques feed IH signals solely at the input layer by employing (a) special delimiter tokens $(S_0, S_1)$ or (b) instruction segment embeddings $(\vec{S}_0, \vec{S}_1)$ that are added to the input token embeddings $\{\vec{T}_1, \vec{T}_2, .., \vec{T}_n\}$. Our proposed approach (c) differs fundamentally by injecting IH signals into every decoder layer, leading to a more robust enforcement of the IH.

**Recipe for a Defense.** Most of these defenses (Wallace et al., 2024; Chen et al., 2024a; Wu et al., 2024; Chen et al., 2024b) follow a common high-level procedure to create robust models, which we outline below.

1. Establish an instruction hierarchy (IH) by defining the number of privilege levels and their relative order of importance (e.g., $P_0 > P_1 > P_2$).
2. Construct an adversarial training dataset $\mathcal{D}'$ comprising examples with conflicting instructions embedded within different parts of the input (analogous to a prompt injection attack).
3. Modify the LLM to accommodate IH signals that encode the privilege levels of each token.
4. Train the modified LLM using $\mathcal{D}'$ to prioritize instructions associated with higher privilege levels.

Existing defenses differ primarily in how they modify the LLM to process IH signals and how they train the LLM (Steps 3 and 4 above). To illustrate, consider a simplified scenario with two privilege levels, $P_0 > P_1$. (Wallace et al., 2024; Chen et al., 2024a) use special delimiter tokens $(S_0, S_1)$ to indicate the privilege levels of input tokens (as depicted in Fig. 2) and train the model using supervised fine-tuning (SFT). *SecAlign* (Chen et al., 2024b) also encodes IH signals using delimiters and trains the model using direct preference optimization (DPO). Another approach, *Instructional Segment Embedding* (ISE) (Wu et al., 2024), proposes adding trainable segment embeddings to the input token embeddings to encode privilege level information.

**Limitation of Existing Defenses.** Our work focuses on the method of injecting the IH signal into the LLM. A common characteristic of prior defenses is that they inject the IH signal exclusively at the input layer, either through special delimiter tokens or by appending segment embeddings to the input token embeddings. However, these input-level IH signals degrade as they propagates through the decoder layers. To demonstrate this, we encode 100 prompts from the AlpacaEval dataset with two different privilege levels and compare the cosine similarity of the intermediate representations across different layers of the Llama-3.2-3B model in Fig. 3. We observe that the similarity between the representations increases as we go deeper into the decoder layers, indicating that the representations may fail to adequately preserve the IH signals. We hypothesize that this limits the effectiveness of the IH signals in enforcing the instruction hierarchy as it propagates through the decoder layers.

## 4 OUR PROPOSAL: AUGMENTED INTERMEDIATE REPRESENTATION

The primary goal of our work is to enhance the efficacy of IH signals by injecting them directly into all layers of the model. We do so by modifying the decoder block to incorporate the IH signal.

**Notations.** Before explaining our proposal, we introduce some notation. Let $\vec{x}_{ij}$ denote the intermediate token representation of the $i^{th}$ input token in the $j^{th}$ decoder block. Assuming that we have $K$ privilege levels, let's use $k_i \in [0, K)$ to denote the privilege level corresponding to the $i^{th}$ token.

**Design.** We set out to find a method for injecting IH signals to each decoder layer in a way that allows the IH signal to be customized to the intermediate representations at the input of each layer. The key changes made by AIR to the decoder block are illustrated in Fig. 4. AIR introduces a trainable embedding table $S_j$ to each decoder block, consisting of $K$ entries - one for each privilege level in the IH (Fig. 4 shows $K = 2$ entries for simplicity). The vectors in this table are sized to have the same dimensionality as the intermediate token representations $\vec{x}_{ij}$. AIR directly injects the IH signals ($k_i$) to all the decoder blocks as shown in Fig. 2c. The injected IH signal is used to index the IH embedding table $S_j$ to retrieve an IH vector, which then augments the intermediate token representation $\vec{x}_{ij}$ to become $\vec{x}'_{ij}$, as defined by:

$$\vec{x}'_{ij} = \vec{x}_{ij} + \vec{s}^{\,k}_j, \quad \text{where } \vec{s}^{\,k}_j = S_j[k_i] \tag{1}$$

We also augment the intermediate token representation after the last decoder layer, before it's fed to the linear layers to output the final logits.

**Overheads.** Our method introduces a small increase in the number of parameters. E.g. for Llama3.1-8B (32 decoder layers and hidden representations of size 4096), with 3 privilege levels, we require a total of $(32+1) \times 3 \times 4096 = 0.4M$ extra parameters (i.e. 0.005% increase). While additional compute is needed to train the model (see Section 5.2), it is similar to the overheads incurred in prior works (Wallace et al., 2024; Chen et al., 2024a;b). The increase in the compute for inference is negligibly small.

**Similarity to Research on Positional Embedding.** Our proposal shares an interesting similarity with the research on positional embeddings. While earlier works primarily injected positional information at the input layer, often in the form of sinusoidal positional encoding (Vaswani et al., 2017) or learnable positional embeddings (Devlin et al., 2019), more recent methods have explored alternative approaches. Notably, Rotary Position Embedding (RoPE) (Su et al., 2024) injects relative positional information directly into the self-attention mechanisms within all layers of the Transformer. Integrating positional information throughout the model's architecture, rather than just at the initial input stage, has been shown to be a significant factor in enhancing the performance of large language models (Su et al., 2024; Zhao et al., 2023; Dufter et al., 2022). Our proposal applies the same underlying principle—distributing critical privilege information across all layers—to improve model security against prompt injection attacks.

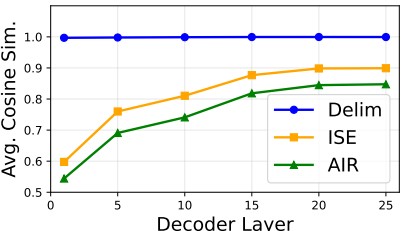

Figure 3: Comparison of average cosine similarity between hidden representations of tokens encoded with different privilege level using different instruction hierarchy injection mechanisms (Delim, ISE, AIR). AIR has lower similarity (better separation) across all layers.

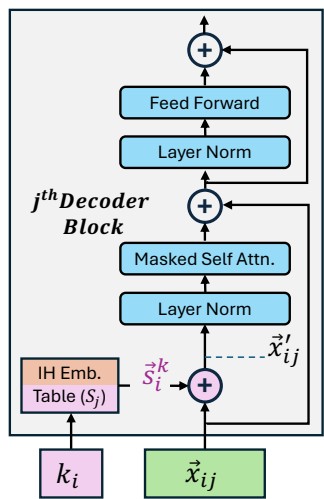

Figure 4: AIR incorporates a trainable embedding table within each decoder block. The information hierarchy signal serves as an index to this table, with the retrieved embedding augmenting the intermediate representation.

## 5 Experimental Setup

Our experimental evaluations aim to quantify the impact of different mechanisms for injecting IH signals on model utility (performance in non-adversarial settings) and robustness (resilience under attack). We describe key details of the experimental setup in this section. Additional details can be found in Appendix B.

### 5.1 Models

We consider three pre-trained base models of varying sizes: Llama-3.2-3B (AI, 2024), Qwen2.5-7B (Team, 2024), and Llama-3.1-8B (Grattafiori et al., 2024). In their original pre-trained state, these

> <|begin_of_text|><|start_header_id|>system<|end_header_id|>Below is an instruction that describes a task, paired with an input that provides further context. Write a response that appropriately completes the request.<|eot_id|><|start_header_id|>user<|end_header_id|>Evaluate this sentence for spelling and grammar mistakes. He finnished his meal and left the resturant<|eot_id|><|start_header_id|>assistant<|end_header_id|>There are two spelling errors in the sentence. The corrected sentence should be: "He finished his meal and left the restaurant."<|eot_id|>

Figure 5: A sample from the Alpaca dataset formatted using a chat template. Each example consists of an instruction $I$, an optional data segment $D$ and the response $R$. We use 3 privilege levels: $P_0 > P_1 > P_2$ to indicate the relative priority of different segments.

models exhibit limited instruction-following capabilities. We adapt the architecture of these models to facilitate the injection of IH signals and subsequently train them as described below.

## 5.2 TRAINING

For a fair comparison, all models in our experiments undergo the same training procedure, regardless of the IH injection mechanism. This procedure involves two sequential rounds of training:

1. **Non-adversarial Instruction Tuning:** First, to instill instruction-following capabilities, the base models undergo full fine-tuning with SFT using an instruction-following dataset. The learning rate (LR) is set to $2 \times 10^{-5}$ for Llama-3.2-3B, and $1 \times 10^{-5}$ for Qwen-2.5-7B and Llama-3.1-8B.
2. **Adversarial Robustness Training:** Subsequently, to enhance robustness against prompt injection attacks, the models undergo a second stage of fine-tuning using a curated adversarial dataset. For this adversarial training stage, we investigate two fine-tuning methodologies:
   - **SFT:** We employ full fine-tuning with a LR of $1 \times 10^{-5}$
   - **DPO:** We perform parameter efficient fine-tuning using LoRA (Hu et al., 2022) with a LR of $2 \times 10^{-4}$.

Each round consists of 3 epochs of training using the AdamW (Loshchilov & Hutter, 2017) optimizer and a linear LR scheduler. Details of the training datasets used for the two rounds are provided in Appendix B.1.

## 5.3 DEFENSES

This subsection details the Instruction Hierarchy (IH) adopted in our experiments and the various mechanisms evaluated for injecting IH signals into the models.

**Instruction Hierarchy (IH).** We define three hierarchical levels of privilege, $P_0 > P_1 > P_2$, as illustrated in Fig. 5. $P_0$ is assigned to system and user instruction tokens. $P_1$ is assigned to tokens within the data segment. $P_2$ is associated with the model's response tokens.

**IH Injection Mechanisms.** In addition to AIR, our proposed approach, we evaluate two existing methods for injecting IH signals:

1. **Delimiters (Wallace et al., 2024; Chen et al., 2024a):** We use two trainable special tokens, *[INST]* and *[INPT]*, to explicitly mark the beginning of instruction (privilege $P_0$) and input (privilege $P_1$) segments, respectively.
2. **Instructional Segment Embedding (ISE) (Wu et al., 2024):** This method adds distinct, trainable embeddings to the token representations to indicate the IH level of each token in the input.

**Connection to Prior Work.** Existing defense strategies can often be characterized by their choice of IH signal injection mechanism and the adversarial robustness training technique employed. For instance, the methods in (Wallace et al., 2024) and (Chen et al., 2024a) can be viewed as utilizing *Delimiters* in conjunction with SFT. The approach in (Wu et al., 2024) employs *ISE* with SFT. *SecAlign* (Chen et al., 2024b) uses *Delimiters* with DPO. Our work extends these investigations by systematically evaluating a broader matrix of IH injection mechanisms (Delimiters, ISE, AIR) and adversarial training techniques (SFT, DPO), including combinations not explored in prior studies.

## 5.4 Evaluation Methodology

Following the training stages, the models are evaluated on two key aspects: utility in non-adversarial settings and robustness against prompt injection attacks. We use two datasets- AlpacaFarm (Dubois et al., 2023) and SEP (Zverev et al., 2024) to measure both utility and robustness.

**AlpacaFarm.** To assess model utility, responses are generated for the 805 test instances from the AlpacaFarm dataset. Each instance in this dataset consists of an instruction and an optional input segment. We employ AlpacaEval 2.0 (Li et al., 2023) for utility evaluation. This framework computes the win rate of the generated responses by comparing them against reference responses using a judge model. For our evaluations, responses from the `text-davinci-003` model serve as the reference and Llama-3-70B-Instruct is used as the judge model. Robustness evaluations are conducted on the subset of 208 instances from the AlpacaFarm dataset that contain input segments ($D$). These input segments are modified to simulate prompt injection attacks by appending an adversarial prefix $D'_p$ and a targeted adversarial instruction $I'$ (i.e., $D \to D + D'_p + I'$). For all robustness tests, the adversarial instruction $I'$ is set to `print exactly 'hacked!'`. The content of the adversarial prefix $D'_p$ varies depending on the attack strategy. We consider two categories of attacks:

- **Static Attacks (Black-Box):** We evaluate against four predefined static attack strategies: *Naive*, *Ignore*, *Completion*, and *Escape Separation*. Attack Success Rate (ASR) is reported across the 208 test instances by determining if the model's response contains the literal phrase `hacked!`.
- **Gradient-Based Attack (White-Box):** We evaluate white-box robustness using two recent, highly effective attacks: momentum-enhanced GCG (Zhang & Wei, 2025) and the Astra Pandya et al. (2025). For both attacks we optimize a 100-token random prefix for 200 (DPO models) or 50 (SFT models) steps to minimize attack loss. For Astra, half of the steps are used to optimize the attention loss (warm-start process) and the rest are used for GCG. ASR is measured using the likelihood (from model's logits) of generating the target phrase `hacked!`.

**SEP.** Zverev et al. (2024) propose a methodology to evaluate a model's ability to separate instructions from data using the SEP dataset. This dataset contains 9160 examples—each comprising an instruction $s_i$, associated data $d_i$, a probe $x_i$, and a witness $w_i$. The probe $x_i$ instructs the model to include the witness $w_i$ in its response. To evaluate utility, the probe is randomly inserted at the beginning or end of the *instruction segment*. The model's response is then checked for the presence of $w_i$. Since the probe is part of the instruction segment, the model's output should ideally contain $w_i$. Utility is therefore measured as the fraction of responses that include the witness. If $\{y_i^I\}_{i=1}^n$ denotes the set of $n$ responses where the probe was inserted into the instruction segment, the *empirical utility score* $U$ is calculated as: $U = \frac{1}{n} \sum_{i=1}^n \mathbb{1}_{\{w_i \in y_i^I\}}$. To evaluate robustness, the probe is similarly inserted randomly at the beginning or end of the *data segment*, and the response is checked for $w_i$. In this case, because the probe is within the data segment, the model should ideally ignore the probe's instruction, and its output should not contain $w_i$. Zverev et al. (2024) propose the *empirical separation score* $S$ to quantify how well the model distinguishes instructions in the instruction segment from those embedded in the data segment. If $\{y_i^D\}_{i=1}^n$ denotes the set of $n$ responses where the probe was inserted into the data segment, the empirical separation score $S$ is calculated as: $S = \frac{\sum_{i=1}^n \mathbb{1}_{\{w_i \in y_i^I \land w_i \notin y_i^D\}}}{\sum_{i=1}^n \mathbb{1}_{\{w_i \in y_i^I\}}}$.

A higher separation score indicates greater robustness against prompt injection attacks.

## 6 Results

### 6.1 AlpacaFarm

**Utility.** Figure 6 compares the utility of models trained with different adversarial training methods (DPO, SFT) and IH injection mechanisms, evaluated on the AlpacaFarm dataset. Compared to a model trained only non-adversarially (*None* in Fig. 6), our proposed AIR method generally does not significantly degrade model utility. At most we observe a $< 2\%$ degradation in utility (for Qwen-2.5-7B trained with DPO).

**Robustness (Static Attacks).** Table 1 provides the ASRs for models with different defenses against *Naive*, *Ignore*, *Completion*, and *Escape Separation* attacks, as well as the SEP benchmark. Although the training and test set examples are distinct, the model encounters the first two attacks are in-distribution as they are seen during adversarial training. We find that all three IH injection mechanisms

Table 1: Attack success rates ↓ (%) for models trained with different IH injection mechanisms (None, Delim, ISE, AIR) and adversarial training techniques (None, SFT, DPO) under various static and gradient-based attacks crafted from the AlpacaFarm dataset. Numbers in **bold** indicate that the corresponding IH mechanism outperforms other methods for a given attack.

| Model | Attack | None | SFT | | | — DPO | | |
|---|---|---|---|---|---|---|---|---|
| | | None | Delim | ISE | AIR | Delim | ISE | AIR |
| Llama-3.2-3B | Naive | 1 | 0.0 | 0.0 | 0.0 | 0.0 | 0.0 | 0.0 |
| | Ignore | 2.5 | 0.0 | 0.0 | 0.0 | 0.0 | 0.0 | 0.0 |
| | Completion | 3.8 | 1 | 0.5 | **0.0** | 0.0 | 0.0 | 0.0 |
| | Escape Sep. | 1.4 | 0.5 | 0.5 | 0.5 | 0.0 | 0.0 | 0.0 |
| | SEP | 17.7 | 4.3 | 3.1 | **2.7** | 2.6 | 2.6 | 2.6 |
| | GCG | 77.5 | 38 | 48.1 | **4.1** | 29.1 | 46.6 | **5.2** |
| | Astra | 54.4 | 14.5 | 25.8 | **0.1** | 34.5 | 57.3 | 23.8 |
| Qwen-2.5-7B | Naive | 3.4 | 0.0 | 0.5 | 0.0 | 0.0 | 0.0 | 0.0 |
| | Ignore | 2.9 | 0.0 | 0.0 | 0.0 | 0.0 | 0.0 | 0.0 |
| | Completion | 3.8 | 1 | 0.0 | 0.0 | 0.0 | 0.0 | 0.0 |
| | Escape Sep. | 2.9 | 0.5 | 0.5 | 0.5 | 0.5 | 0.0 | 0.0 |
| | SEP | 41.6 | 4.9 | 3.7 | **3.0** | 4.4 | 4.8 | **3.4** |
| | GCG | 99.5 | 88 | 36.6 | **22.6** | 32 | 7.7 | **1.6** |
| | Astra | 99.4 | 69.0 | 39.2 | **2.4** | 19.9 | 2.3 | **0.9** |
| Llama-3.1-8B | Naive | 0.5 | 0.0 | 0.0 | 0.0 | 0.0 | 0.0 | 0.0 |
| | Ignore | 2.5 | 0.0 | 0.0 | 0.0 | 0.0 | 0.0 | 0.0 |
| | Completion | 3.8 | 0.0 | 0.0 | 0.0 | 0.0 | 0.0 | 0.0 |
| | Escape Sep. | 1.4 | 0.5 | 0.0 | 0.0 | 0.0 | 0.0 | 0.0 |
| | SEP | 33.7 | 5.3 | 3.1 | 3.1 | 3.9 | 2.8 | **2.2** |
| | GCG | 99.5 | 77 | 19.9 | **11.3** | 13 | 4 | **2.8** |
| | Astra | 97.9 | 76.3 | 0.2 | **0.1** | 36.9 | 1.2 | **1.0** |

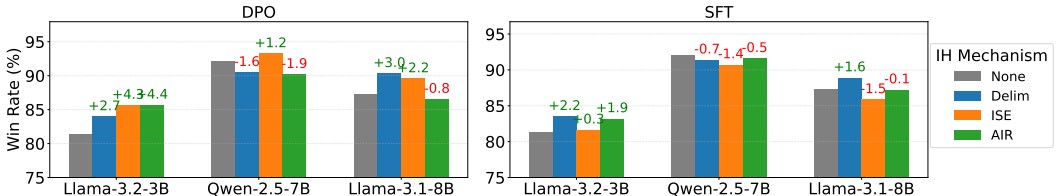

Figure 6: Comparison of win rates for models trained with different IH injection mechanisms. In most cases, the Win Rate of the model trained with IH is comparable to that of the baseline win rate of a non-adversarially trained model with no IH signals (indicated by *None*).

(*Delimiter*, *ISE*, and *AIR*) offer near-perfect protection against the first four attacks. For SEP, we find that AIR offers equal or better protection compared to other methods for all models.

**Robustness (Gradient-Based Attack).** Figure 7 illustrates the comparative performance of these defenses against the Momentum-Boosted GCG attack. The figure plots the attacker's loss—calculated relative to the target adversarial response—as a function of GCG optimization steps. Each line indicates the mean loss over 208 test instances, with shaded regions representing the standard deviation. Results are presented separately for models adversarially trained with DPO (first row of plots) and SFT (second row). As anticipated, the attacker's loss diminishes with more GCG optimization steps, signifying increased attack efficacy. Notably, models defended by our proposed AIR mechanism consistently incur a significantly higher average attacker loss compared to those defended by *ISE* or *Delimiters*. Furthermore, GCG's ASR (GCG in Table 1) against AIR is **1.6× to 9.2× lower** compared to next best defense, underscoring AIR's superior robustness. Our findings also reveal that adversarial training with DPO yields more robust models than SFT, corroborating results from SecAlign (Chen et al., 2024b). We observe similar trends for the Astra attack. Astra's ASR (Astra in Table 1) against AIR is up to **145× lower** for SFT and **2.5× lower** for DPO compared to the next best defense. A detailed discussion of the results from Astra is presented in Appendix C.

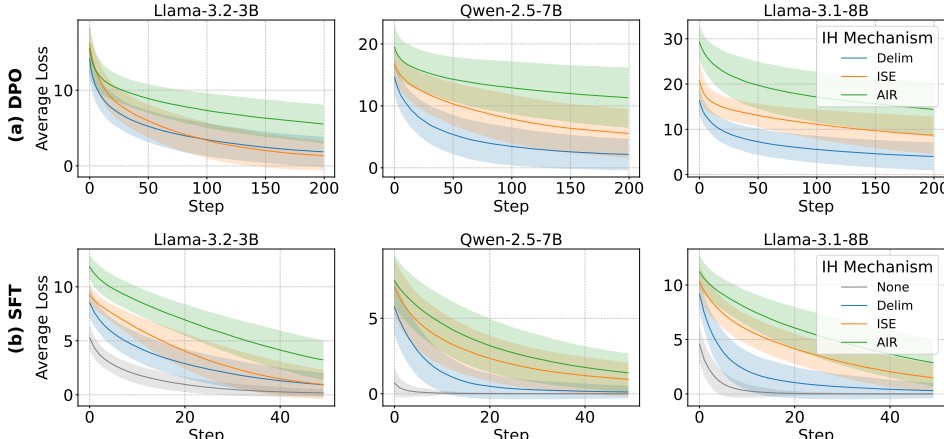

Figure 7: Average loss from the Momentum-Boosted GCG attack comparing different defenses during various points in the optimization process. AIR is more robust to GCG with a higher average loss compared to prior works across all models and both optimization methods.

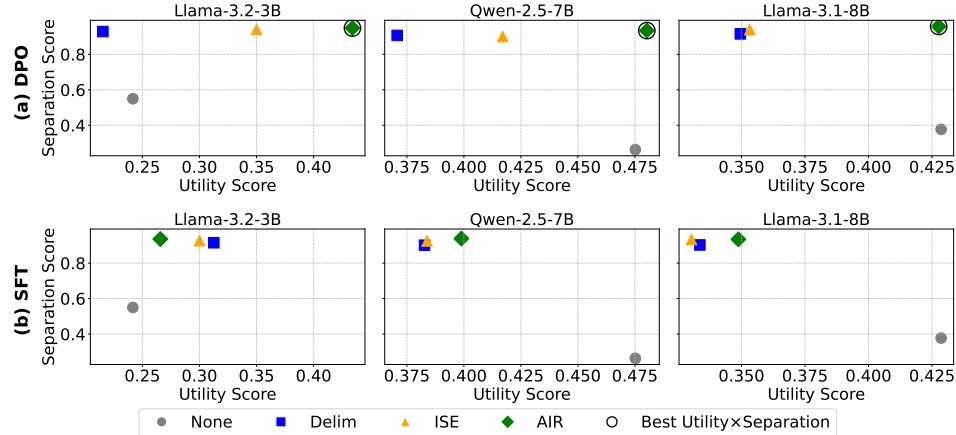

Figure 8: Utility and Separation scores derived from the SEP dataset. IH mechanisms with the best $utility \times separation$ for each model (across both DPO and SFT) are marked with ◯

## 6.2 SEP

Figure 8 plots empirical separation and utility scores, comparing the different IH injection mechanisms. For models trained with DPO (Fig. 8a), AIR achieves the highest separation and utility scores, outperforming other IH mechanisms as well as all models trained with SFT in these combined metrics. For models trained with SFT, AIR maintains higher separation scores than other methods across all models. However, in some instances (e.g., Qwen-2.5-7B, Llama-3.1-8B), AIR-SFT's utility can be lower than the *None* baseline (which undergoes only non-adversarial training). Overall, these results indicate that AIR consistently enhances the model's ability to separate data from instructions and, when trained with DPO, provides the best utility-separation tradeoff for the evaluated models.

## 7 CONCLUSION

Our paper proposes a new defense for prompt injection attacks. We study the various mechanisms of injecting instruction hierarchy information in prior work and find that they suffer from a crucial limitation – they only insert the IH information to the input layer of the LLM, which limits the efficacy of the IH signal. To overcome this drawback, we propose Augmented Intermediate Representations (AIR), which injects the IH signals into all the decoder layers in the model. Through extensive empirical studies on models of different sizes (3B, 7B, 8B), and training techniques (SFT, DPO), we show that our proposal can improve robustness against gradient-based attacks by $1.6\times$ to $9.2\times$, without significant degradation in utility.

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

## A  LIMITATIONS AND FUTURE WORK

While our defense demonstrates strong average resilience to white-box attacks, it does not provide formal robustness guarantees, meaning specific outliers or advanced attacks might still succeed. This is a common limitation in the current LLM robustness research landscape. Additionally, our utility and robustness evaluations, similar to prior work, are confined to single-turn interactions using the AlpacaFarm and SEP datasets. Evaluating our proposal's effectiveness in multi-turn conversational settings and complex agentic workflows is therefore a key direction for future work.

## B  ADDITIONAL EXPERIMENTAL DETAILS

### B.1  TRAINING DATASETS

**Non-Adversarial Dataset.** For the first stage of training (*non-adversarial instruction tuning*), we employed the cleaned version (Ruebsamen, 2024) of the Alpaca dataset (Taori et al., 2023). This dataset comprises approximately 52K examples. As illustrated in Fig. 5, each example typically consists of an instruction (I), an optional input segment (D), and the desired response (R). The models are trained to generate R given I and D (when present), formatted according to a specific chat template.

For the second stage, *adversarial robustness training*, we constructed two distinct adversarial versions of the Alpaca dataset: one for SFT and another for DPO.

**Adversarial SFT Dataset.** This dataset incorporates all examples from the original Alpaca dataset.

- Examples that originally lack an input segment ($D$) are included unmodified.
- For examples that do contain an input segment ($D$), half are included unmodified. The other half are modified to simulate a prompt injection attack. The input segment $D$ is transformed into $\hat{D}$ by concatenating the original input, an adversarial prefix $D'_p$, and an adversarial instruction $I'$ (i.e., $\hat{D} = D + D'_p + I'$). The adversarial prefix $D'_p$ is determined by either the *Naive* or *Ignore* attack strategy, chosen with uniform probability. The adversarial instruction $I'$ is an instruction randomly selected from a different example within the Alpaca dataset.

This adversarial SFT dataset can be represented as collections of tuples $(I, \bar{D}, R)$, where $\bar{D}$ is either the original input $D$, the modified input $\hat{D}$, or absent (if the original example had no input segment).

**Adversarial DPO Dataset.** To construct the preference dataset for DPO, we exclusively used Alpaca examples that contain an input segment ($D$). For each such example, we generated a corrupted input segment $\hat{D}$ using the same *Naive* or *Ignore* prompt injection techniques (resulting in $\hat{D} = D + D'_p + I'$ as described above). The preference pair consists of the original instruction $I$ and the corrupted input $\hat{D}$. The chosen response is the original, correct response $R$ from the Alpaca dataset (corresponding to $I$ and $D$). The rejected response is the response $R'$ associated with the adversarial instruction $I'$ in its original Alpaca example. This DPO dataset is a collection of tuples $(I, \hat{D}, R, R')$.

All examples across these datasets were formatted using the chat template depicted in Fig. 5 before being used to train the models.

### B.2  MODEL AND TRAINING CONFIGURATIONS

For all training runs, we use a batch size of 4 with 4 steps of gradient accumulation for both rounds of training. We employed Parameter-Efficient Fine-Tuning (PEFT) using the Low-Rank Adaptation (LoRA) technique to fine-tune the model with DPO. Specifically, we fine-tuned the query (`q_proj`) and value (`v_proj`) projection layers. The LoRA hyperparameters were set with a rank ($r = 64$), `lora_alpha`= 8, and `lora_dropout`= 0.1.

**Embedding Table Initialization.** Our method introduces embedding tables within the decoder block to augment intermediate representations. These tables are initialized by default with vectors sampled from a normal distribution with a standard deviation of 0.02 ($\mathcal{N}(0, 0.02^2)$). While this initialization proved effective for Llama models, it yielded suboptimal robustness performance for

the Qwen model. We attribute this discrepancy to the significantly larger magnitude of intermediate representations produced by Qwen; the default, smaller embedding vectors failed to sufficiently modify these representations. To rectify this, we increased the initialization standard deviation fivefold to $0.1$ ($\mathcal{N}(0, 0.1^2)$) specifically for the Qwen model, which demonstrably improved our defense's effectiveness. For a fair comparison, this same adjusted initialization was applied to the ISE technique when used with Qwen. Due to computational constraints, exhaustive tuning of this hyperparameter was not feasible and is deferred to future work. However, we suggest the following practical guidelines to help with the choice of $\sigma$ for extending our proposal to new models.

*Analyze Activation Scale:* Before training, run a few forward passes on a sample of data (e.g., 100 examples from Alpaca) to measure the average magnitude (L2 norm) of the intermediate representations ($\vec{x}_{ij}$) that AIR will augment.

*Scale Initialization Accordingly:* Use a baseline model (e.g., Llama-3.1-8B with $\sigma = 0.02$) and scale the initialization $\sigma$ for the new model's AIR embedding tables proportionally to its observed activation magnitude.

## C    ADDITIONAL RESULTS

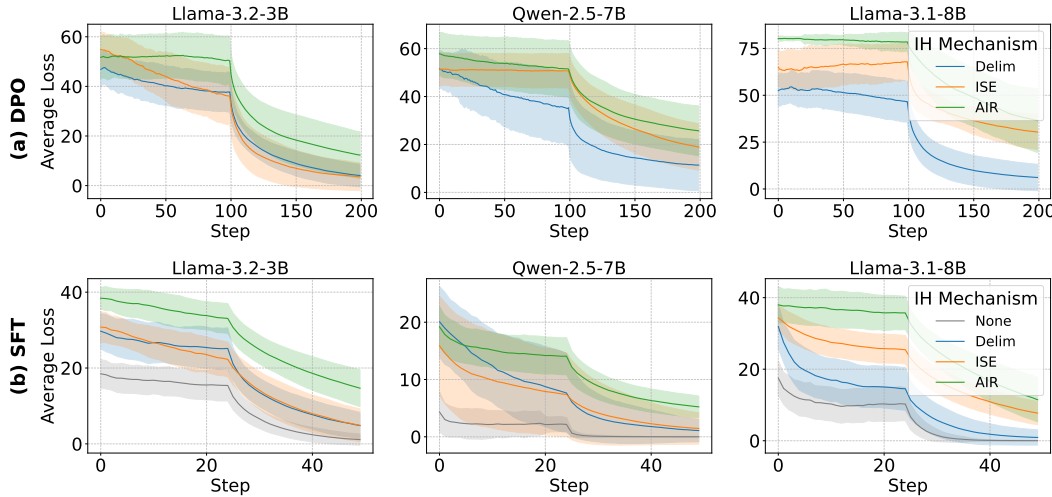

Figure 9: Average loss from the Astra attack comparing different defenses during various points in the optimization process. AIR is more robust to Astra with a higher average loss compared to prior works across all models and both optimization methods.

**Astra Attack.** The Astra attack (Pandya et al., 2025) has two phases. In the first phase, the attack optimizes the adversarial prefix to minimize the attention loss. Doing so focus the model's attention on the adversary's instructions. The prefix found from phase-1 is used as the starting point for the GCG attack in phase-2. We refer the reader to Section 5 of the Astra paper (Pandya et al., 2025) for a description of the attention loss. The loss curves for this attack are shown in Fig. 9. Note that we use half of the optimization steps for phase-1 and switch to GCG for phase-2. This switch causes the drop in the loss mid-way through the optimization process. Our results show that across both DPO and SFT, AIR continues to have better robustness (higher adversarial loss) compared to both Delimiter and ISE.

**Progression in Robustness.** The results for GCG and Astra highlight a clear progression in defense efficacy. Recall that the *Delimiters* mechanism injects IH signals via special tokens at segment boundaries, while the *ISE* method applies IH signals (through dedicated embeddings) to all tokens in the input. The enhanced robustness observed when moving from *Delimiters* to *ISE* suggests the benefit of more pervasive IH signal application at the input level. Our AIR approach further advances this principle; by injecting IH signals directly into all decoder layers, rather than confining them to the input representations, AIR achieves a more deeply integrated hierarchical understanding within the model, leading to the observed superior robustness against this strong gradient-based attack.

## D  ADDITIONAL RELATED WORK

**Detection-Based Defenses.** The related work in Section 3.2 primarily discussed defenses designed to enhance model robustness against prompt injection by defining an instruction hierarchy. In addition to these, a significant class of defenses focuses on *detecting* malicious or unintended instructions within user inputs or data segments before they cause the main LLM to deviate from its intended behavior. The core idea is to employ a detection mechanism as a preliminary check or ongoing monitor. Several approaches to detection-based defenses have been proposed:

- **LLM-Powered Detectors:** A common strategy is to leverage an LLM itself as a detector. These approaches include using zero-shot or few-shot prompting of an LLM to ascertain if an input contains hidden or malicious instructions (Stuart Armstrong, 2022). Another technique involves fine-tuning a dedicated LLM to act as a specialized classifier or "guard" model for identifying malicious prompts or instruction injections (Sharma et al., 2025). Furthermore, LLM self-evaluation techniques have been explored, where the model attempts to determine if it is being manipulated.
- **Known Answer Detection:** Another interesting line of work focuses on testing if the LLM returns a known answer in the presence of potentially malicious tokens (Yohei, 2022). This method uses a special instruction where the answer is only known to the detector. If the response fails to provide the expected answer in the presence of a data segment, then the data segment is flagged as containing a prompt injection attack. A recent work (Liu et al., 2025) extends this idea using a game-theoretic foundation to train a detector LLM that is very sensitive to prompt injection attacks, achieving near-perfect scores on benchmarks. However, such defenses remain vulnerable to adaptive attacks (e.g., if the attacker instructs the LLM to return the known answer before following the attacker's instructions).
- **Output Analysis and Verification:** Instead of, or in addition to, input checks, some defenses analyze the LLM's output. This includes response checking, which evaluates whether the LLM's output aligns with the intended task or original user instruction, where deviations might indicate manipulation (Sharma et al., 2025). Perplexity-based detection has also been explored to identify anomalous outputs (Alon & Kamfonas, 2023).

While detection-based methods offer a valuable layer of security, they remain vulnerable to adaptive attacks. Therefore, such defenses can complement our proposed defense, which is designed to make the model inherently robust to prompt injection attacks.

**ASIDE.** A concurrent work, ASIDE (Zverev et al., 2025) also identifies the "IH signal degradation" issue in methods like ISE. While our AIR approach addresses this by injecting IH embeddings into every layer, ASIDE proposes an alternative: enforcing an orthogonality constraint on the input-layer embeddings. This is designed to make the IH signal (e.g., privilege levels) and the token/positional information independent, thereby improving the signal's persistence as it propagates through the network. This highlights a shared recognition of the problem, with distinct architectural solutions.

## E  ADDITIONAL INTERPRETABILITY EXPERIMENTS

To further validate our hypothesis from Section 3.2—that input-only Instruction Hierarchy (IH) signals degrade or are not well-represented—we conducted a linear probing experiment. We measure how linearly separable the intermediate representations of tokens are based on their assigned privilege level.

**Methodology:** For 100 prompts from the AlpacaEval dataset, we collect the intermediate representations from the output of each decoder block in Llama3.2-3B for all tokens. Each token is labeled with its privilege level ($P_0$ or $P_1$). For each layer, we then train a simple linear classifier (probe) to predict the privilege level (0 or 1) from the token's representation at that layer.

**Results:** We evaluate our probes on a held-out test set of 100 different prompts. The results are shown in Fig. 10.

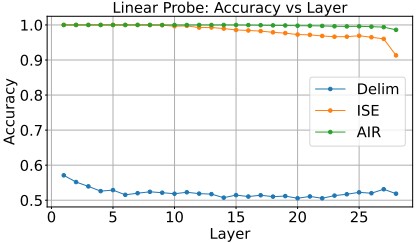

Figure 10: Comparison of linear probe accuracy across different instruction hierarchy injection mechanisms. Intermediate representations produced by AIR has better linear separability compared to other methods across all layers.

*Delimiters (Delim):* The probe's accuracy is near chance ($\approx 50\% - 55\%$) across all layers. This strongly suggests the privilege information is not linearly encoded in the token representations, forcing the model to rely on a different, less robust mechanism.

*Instruction Segment Embedding (ISE):* The probe achieves perfect accuracy in the initial layers, but this visibly degrades as representations propagate, dropping to $91\%$ by the final layer. This directly confirms our hypothesis that input-only signals lose distinctiveness.

*Augmented Intermediate Representations (AIR):* In contrast, the probe for AIR maintains near-perfect accuracy across all layers. This provides clear empirical evidence that AIR successfully injects a persistent, robust, and linearly separable IH signal at every processing stage.

## F  ADDITIONAL STATIC ATTACKS

To further validate the improved robustness of AIR compared to prior works, we provide the attack success rates on the indirect prompt injection attacks from the BIPIA dataset in Table 2. We report numbers of three different tasks in this dataset: email, code and table. We restrict evaluations to samples that can be judged programatically (without using LLM as a judge). Consistent with our results on other benchmarks, we find that AIR exhibits higher robustness to attacks across all tasks compared to ISE and Delim methods.

Table 2: Attack success rates on different tasks of the BIPIA dataset

| Model | Task | None | SFT | | | DPO | | |
|-------|------|------|------|-----|-----|------|-----|-----|
| | | | Delim | ISE | AIR | Delim | ISE | AIR |
| llama3.2-3b | email | 46 | 3.26 | 2.86 | **1.13** | **0** | **0** | **0** |
| | code | 21.76 | 12.57 | 12.1 | **7.8** | 4.3 | 2.4 | **0.85** |
| | table | 58.4 | 31.7 | 16.7 | 13 | 2.86 | **0.2** | **0.2** |
| llama3.1-8b | email | 77.13 | 11 | 7.7 | **4.7** | **0** | 0.266 | **0** |
| | code | 17.7 | 5.69 | 15.3 | **5.04** | 0.18 | 1.57 | **0.13** |
| | table | 90.2 | 27.8 | 12.73 | **8.3** | 1.36 | 1.06 | **0.6** |
| qwen2.5-7b | email | 70.26 | 18.2 | 13.2 | **11.8** | 11.13 | 10.67 | **6.4** |
| | code | 12 | 0.21 | 0.45 | **0.12** | **0** | **0** | **0** |
| | table | 76.53 | 37.43 | **28.3** | 30.7 | 34.53 | 16.9 | **12.1** |

## G  ADDITIONAL UTILITY MEASUREMENTS

In addition to the utility measures (SEP, AlpacaEval) provided in the main paper, we provide the MMLU scores for the models trained with different IH signals in Fig. 11. The results show that models trained with AIR perform comparably to ones trained with Delim and ISE across all architectures and training methods.

## H  COMPUTE RESOURCES

We use compute nodes with $8\times$ A100 GPUs paired with 256 CPU cores and 1TB of memory and 25 TB of storage for all our experiments. Note that most of our training runs complete within 2 hrs. The gradient based attacks need more time due to their sequential nature and require around 30 mins per example with a single gpu.

## I  LLM USAGE

We used an LLM to assist in the preparation of this manuscript, primarily to improve the clarity, grammar, and succinctness of the text. All the model's suggestions were carefully reviewed by the

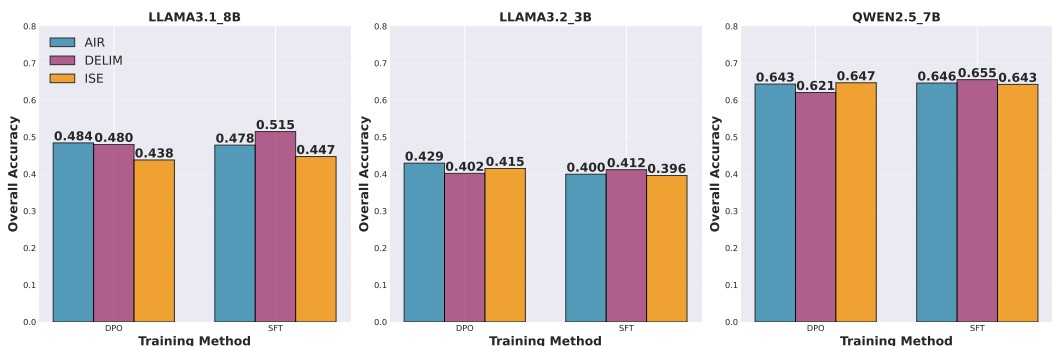

Figure 11: Comparison of MMLU accuracy across models trained with different IH injection methods.

authors. The core scientific ideas, methodology, and conclusions presented are solely the work of the authors.

## J  SOCIETAL IMPACT

The research presented in this paper aims to enhance the security and reliability of LLMs by proposing a more robust defense (AIR) against prompt injection attacks. Positive impacts include increased user trust and safety when interacting with LLM-powered applications, particularly those processing untrusted external data like emails or web content. By making models less susceptible to malicious instruction hijacking, this work could facilitate the safer deployment of helpful AI agents in various domains, reduce the potential for AI-driven misinformation or data exfiltration triggered by such attacks, and contribute to the broader adoption of LLMs for beneficial tasks. However, potential negative consequences or challenges must also be considered. Improved defenses might lead to over-reliance or a false sense of complete security, potentially discouraging complementary security measures. Ultimately, while techniques like AIR contribute positively towards trustworthy AI, they should be viewed as one component within a larger framework for responsible AI development and deployment.

## K  REPRODUCIBILITY STATEMENT

The code to reproduce our results are included in the supplementary material. Key experimental details are provided in Section 5 and Appendix B.

