# OpenReview forum: "Enforcing Instruction Hierarchy via Augmented Intermediate Representations"
_ICLR.cc/2026/Conference — Submitted to ICLR 2026_

### Official Review · Reviewer_Pn8n · 2025-10-30

**Soundness:** 3
**Presentation:** 3
**Contribution:** 3
**Rating:** 6
**Confidence:** 4

**Summary:**

This paper proposes Augmented Intermediate Representations (AIR), a defense mechanism against prompt injection attacks in LLMs. Existing defenses introduce an Instruction Hierarchy (IH) signal, indicating the privilege level of input tokens, but inject it only at the input layer. The authors hypothesize that this limits the model’s ability to maintain hierarchical distinctions as information propagates through layers. To address this, AIR injects layer-specific, trainable embeddings encoding IH signals into every decoder block of the model. This recurrent injection ensures that privilege information remains accessible throughout the network.

**Strengths:**

1.	The paper proposes a new architectural idea that injects Instruction Hierarchy signals at every layer of the model rather than only at the input, offering a fresh approach to prompt-injection defense distinct from earlier input-based methods.
2.	The experiments cover different model sizes and both SFT and DPO training setups, and consistently show gains in robustness.
3.	The problem addressed, protecting models from prompt injection in settings involving untrusted data and agent workflows, is timely and highly important.

**Weaknesses:**

1.	Utility is measured mostly via AlpacaEval win rates. There is no assessment of factual accuracy or reasoning (e.g., MMLU), so it is difficult to judge whether AIR affects model quality in benign settings. Including standard benchmarks would strengthen the claims.
2.	Although the paper suggests AIR can be applied to direct prompt attacks and agent settings, no experiments verify this. Multi-turn, retrieval-augmented, and user-as-attacker (jailbreak) scenarios remain unexplored.
3.	There is no visualization of how AIR changes attention patterns across layers. Such analysis could clarify whether AIR genuinely preserves hierarchical separation or merely adds regularization noise.
4.	The SFT models are fully fine-tuned, while DPO models use LoRA. This mismatch may explain the utility drop seen in Figure 8. A controlled comparison (full-FT vs LoRA in both settings) would clarify this.

**Questions:**

1.	Qwen required a much larger initialization scale for the IH embeddings. Can the authors provide a systematic study of initialization and stability? Otherwise, AIR’s robustness may depend on model-specific tuning rather than a generalizable method.
2.	Utility is measured mainly via AlpacaEval win rate. Can the authors report additional evaluations (e.g., MMLU, BLEU, factual consistency, human preference) to confirm that AIR does not subtly degrade model quality in benign settings?
3.	Since the paper reimplements prior defenses, can the authors verify that the reproduced baselines match the original papers’ reported performance, or provide an error margin? This would ensure a fair comparison and strengthen the empirical claims.
4.	The method uses trainable embeddings for privilege levels. Could the authors verify that learning these embeddings is necessary? For example, how does AIR perform if fixed random vectors are used instead? A comparison would clarify whether the improvement comes from the learned hierarchy signal or simply from adding noise/perturbations at each layer.

---

> ### Author Response · Authors · 2025-11-21
> **Rebuttal**
>
> We appreciate the reviewer's insightful comments. To address the concerns regarding robustness validation and parameter sensitivity, we have performed new experiments including MMLU utility measurements and random-vector ablations. Our detailed response is provided below.
>
>
> **1. Sensitivity to initialization of embedding tables in AIR**
>
> The performance of AIR is sensitive to the initialization $\sigma$ primarily if the value is too low relative to the magnitude of the intermediate representations ($\vec{x}_{ij}$) it is being added to. Our investigation revealed exactly this: the magnitude of the Qwen model's intermediate activations is approximately 5x larger (geometric mean of L2 norm across all layers) than that of the Llama models (measured across 100 samples from the Alpaca dataset). Therefore, the default $\sigma=0.02$ (effective for Llama) resulted in embedding vectors that were too small to sufficiently modify Qwen's representations, leading to suboptimal robustness. The 5x increase to $\sigma=0.1$ was chosen as the correction to match the activation scale of the target model, which demonstrably improved effectiveness.
>
> **Practical guidelines for choosing \sigma**
> While a thorough investigation of the sensitivity of this hyperparameter for different models and layers would be interesting, this requires dozens of training runs. While we lack the resources to conduct this study, we suggest the following practical guidelines to help with the choice of $\sigma$.
>
> **Analyze Activation Scale:** Before training, run a few forward passes on a sample of data (e.g., 100 examples from Alpaca) to measure the average magnitude (L2 norm) of the intermediate representations ($\vec{x}_{ij}$) that AIR will augment.
>
> **Scale Initialization Accordingly:** Use a baseline model (e.g., Llama-3.1-8B with $\sigma=0.02$) and scale the initialization $\sigma$ for the new model's AIR embedding tables proportionally to its observed activation magnitude.
>
> We have added the above guidelines to Appendix B.2
>
> **2. Additional utility measures**
> We have conducted additional utility measurements using MMLU. Results are provided in Appendix  G of the revised manuscript. Our results show that models trained with AIR have a utility comparable to baseline models trained without any IH signals.
>
> **3. Error margins**
>
> Unfortunately, there are differences in the training and evaluation setup of prior works, which makes it hard to directly compare reported numbers. However, our experimental setup ensures that the training/attack setup between different defenses is identical, ensuring a fair comparison.
>
> Regarding error margins, we confirm that we have provided these for our primary results. As the reviewer requested, the standard deviation across test instances is visualized as shaded regions in Figure 7 (GCG) and Figure 9 (Astra) to clearly demonstrate statistical significance.
>
> **4. Is learning the embeddings necessary?**
>
> To rule out the possibility that robustness could arise from simply adding noise/perturbations at each layer, we replace the learned embeddings of the Llama3.1-8B AIR model with fixed random vectors and attack it using GCG. We find that the attack success rate jumps from 2.8% to over 99%. This confirms that the model’s robustness indeed comes from the learned instruction hierarchy signals.

---

> > ### Comment · Reviewer_Pn8n · 2025-11-27
> >
> > The authors have addressed some of my concerns. I will keep my score, as the novelty remains limited.

---

### Official Review · Reviewer_kjgc · 2025-10-31

**Soundness:** 2
**Presentation:** 3
**Contribution:** 3
**Rating:** 4
**Confidence:** 4

**Summary:**

The paper introduces a new defense against indirect prompt injections, a type of attack where the attacker injects malicious content into the input context. The authors propose Augmented Intermediate Representations (AIR), which, unlike prior approaches that apply Instruction Hierarchy (IH) information only at the input of the transformer (e.g. using delimiter tokens or IH embeddings), adds IH embeddings at every transformer block. They hypothesize that previous methods lose IH signal strength as it propagates through model layers, which motivates their method. The paper benchmarks AIR on the AlpacaFarm and SEP datasets using Llama-3.2-3B, Qwen-2.5-7B, and Llama-3.1-8B, trained with either SFT or DPO, under both static and gradient-based attacks. On these benchmarks, AIR achieves favorable results.

**Strengths:**

- **Relevant topic**: The paper addresses a relevant, timely, and practical problem.
- **Clear motivation and presentation**: The paper is well-written and clearly explains why the method is needed.
- **Simple approach**: The method is easy to implement and introduces only a relatively small number of additional parameters.
- **Strong empirical section**: The evaluation is extensive and covers multiple base models, attack types, and training regimes (SFT, DPO). See Weaknesses for comments regarding datasets.

**Weaknesses:**

- **Validation of motivating hypothesis**: The hypothesis requires stronger validation. It remains unclear whether existing methods fail due to IH signal degradation, as claimed in section "Limitations of Existing Defenses". Measuring cosine similarity across layers is not sufficient, especially for delimiter-based methods. A simple linear probing experiment (as done e.g. in ASIDE) to test IH separability would strengthen the claim substantially. This is particularly important since identifying this limitation of input-only methods is listed as one of the papers three main contributions.

- **Static attacks fail**: The reported improvements come mainly from gradient-based attacks. Static attacks appear to fail even for the naive baseline, so robustness improvements there seem less meaningful. Including more difficult or diverse attack benchmarks would make the results more convincing.

**Questions:**

- The authors may want to look into ASIDE, which proposes a closely related defense method, also addressing a similar “IH signal degradation” issue in ISE. While not being obligatory, a comparison between AIR and ASIDE would be scientifically valuable, as they both target a similar goal but try to achieve it with different methods: ASIDE enforces IH separation via orthogonal rotations at the input layer, whereas AIR reinforces the IH signal throughout the network with IH embeddings.
- L135 Spelling error: "Ig nore"
- Figure 4 seems to be wrong as a single decoder block contains two masked self-attention computations.

 **References**:
- Zverev et al. “ASIDE: Architectural Separation of Instructions and Data in Language Models.” ICLR 2025 Building Trust in LLMs and LLM Applications Workshop (non-archival), 2025.

---

> ### Author Response · Authors · 2025-11-21
> **Rebuttal**
>
> We thank the reviewer for their comments and suggestions. In response to the feedback, we have added additional experiments to our manuscript. We provide detailed a response to the reviewer's concerns below.
>
> **1. Validation of motivating hypothesis**
>
> Following the reviewer’s suggestion, we ran the linear probe experiment. We have added the full methodology and findings as a new section in the appendix (Appendix E).
> These new results further support our core hypothesis:
>
> **For Delimiters (Delim):** The linear probe's accuracy is at chance-level ($\approx 50\%$). This shows the IH signal is not linearly separable in the token representations at all.
>
> **For Input-Only Signals (ISE):** The probe's accuracy starts perfectly high (1.0) but then visibly degrades in deeper layers to $91\%$, confirming our signal degradation hypothesis.
>
> **For Our Method (AIR):** The probe for AIR maintains near-perfect accuracy across all decoder layers, demonstrating a persistent and separable signal.
>
> This new experiment provides the stronger validation the reviewer requested and supports our paper's contribution.
>
> **2. Additional attacks**
>
> Upon the reviewer’s suggestion, we additionally evaluate AIR on the BIPIA benchmark consisting of indirect prompt injection attacks from three categories (code, table, email). Results are provided in Appendix  F of the revised manuscript. The results show that AIR provides the lowest attack success rate across most tasks compared to ISE and Delim methods.
>
> **3. ASIDE**
>
> We thank the reviewer for highlighting ASIDE. This appears to be concurrent work, and we agree it addresses the same core "IH signal degradation" problem, albeit with a different mechanism (input-layer rotations vs. our all-layer embeddings). We have added a discussion of ASIDE to our additional related work section (Appendix D) to situate our contribution and acknowledge this parallel research.
>
> **4. Typos, Figure 4**
>
> We thank the reviewer for identifying these errors. The error in Figure 4 was particularly egregious. We have fixed these issues in the updated version of the paper.
>
> **References**
>
> [1] Yi, Jingwei, et al. "Benchmarking and defending against indirect prompt injection attacks on large language models." Proceedings of the 31st ACM SIGKDD Conference on Knowledge Discovery and Data Mining V. 1. 2025.

---

### Official Review · Reviewer_PM5M · 2025-11-01

**Soundness:** 3
**Presentation:** 3
**Contribution:** 3
**Rating:** 4
**Confidence:** 3

**Summary:**

This paper addresses prompt injection attacks in LLMs by proposing Augmented Intermediate Representations (AIR), a defense mechanism that injects instruction hierarchy (IH) signals across all decoder layers rather than only at the input layer. The authors argue that existing defenses (using delimiters or input segment embeddings) suffer from signal degradation through the network. Experiments across three models show AIR reduces attack success rates by 1.6-9.2× on gradient-based attacks compared to

**Strengths:**

1. Clear motivation and well-identified limitation: The observation that IH signals degrade through decoder layers (Figure 3) is compelling and provides solid motivation for the proposed approach. The parallel to positional embeddings (RoPE) is insightful.

2. Comprehensive experimental evaluation: The paper evaluates multiple models (3B, 7B, 8B parameters), training methods (SFT, DPO), and attack types (static and gradient-based), demonstrating thoroughness.

3. Minimal overhead: The additional parameters (0.005% for Llama3.1-8B) and inference compute are negligible, making the approach practical.

4. Well-structured presentation: The paper is clearly written with good use of figures and tables to convey results.

**Weaknesses:**

1. Limited theoretical justification: While Figure 3 shows cosine similarity increases across layers, this alone doesn't conclusively prove that IH signal degradation is the limiting factor. Alternative explanations could include:
- The difficulty of learning from input-only signals during training
- The specific architecture's tendency to homogenize representations
- The paper would benefit from ablation studies showing what happens with AIR at only some layers, or from analyzing attention patterns to demonstrate that AIR helps maintain privilege distinctions.

2. Inconsistent performance across training methods: AIR-SFT sometimes shows lower utility than the None baseline (Figure 8b), particularly for Qwen-2.5-7B and Llama-3.1-8B. This is concerning and inadequately explained. The paper should:
- Investigate why this degradation occurs specifically with SFT
- Provide guidance on when to use DPO vs. SFT with AIR
- Discuss potential mitigation strategies

3. Model-specific hyperparameter sensitivity: The need for different initialization strategies for Qwen (σ=0.1 vs. σ=0.02 for Llama) raises concerns about generalization:
- How sensitive is performance to this choice?
- What guidance can be provided for applying AIR to new model families?
- The lack of hyperparameter tuning "due to computational constraints" is unsatisfying for a defense mechanism intended for practical deployment.

4. Limited evaluation scope:
- Only single-turn interactions are tested (acknowledged in limitations)
- No evaluation on real-world prompt injection scenarios beyond AlpacaFarm and SEP
- No comparison with detection-based defenses mentioned in Appendix D
- Static attack evaluation less informative: Since all three IH mechanisms achieve near-perfect defense against static attacks (Table 1), these results don't effectively differentiate approaches. More emphasis should be placed on adaptive attacks.

**Questions:**

Please refer to weaknesses.

---

> ### Author Response · Authors · 2025-11-21
> **Rebuttal**
>
> We thank the reviewer for their feedback. To address the weaknesses and questions raised, we have conducted additional experiments and provide a detailed response below
>
> **1. Theoretical justification showing signal degradation**
>
> We agree that Figure 3, which shows the increasing cosine similarity, serves as strong motivation for our hypothesis rather than a conclusive, formal proof. The alternative explanations provided by the reviewer are not only valid but are, in fact, complementary to our core argument.
>
> **"Difficulty of Learning" and "Homogenization":** We posit that it is precisely because of the Transformer's architectural tendency to homogenize representations (as the reviewer notes) that it becomes "difficult" to train the model to propagate an input-only signal effectively. The input-level signal must be preserved across dozens of processing layers, which is a significant learning burden. The degradation shown in Figure 3 is a symptom of this. Our AIR method is designed to directly alleviate this learning difficulty. By "recurrently" injecting the IH signal at every layer, we remove the burden on the model to learn to propagate this crucial information. The signal is consistently available at every processing stage, acting as a direct counter-force to the natural homogenization.
>
> **Layer Ablation:** Our main results already provide a strong macro-level ablation:
>
> Delimiters: A sparse, input-only signal.
>
> ISE: A dense, input-only signal.
>
> AIR: A dense, all-layer signal. The clear progression in robustness from Delimiters to ISE, and then the significant leap to AIR (as shown in Table 1 and Figure 7), strongly supports our hypothesis that more pervasive, multi-layer signal injection is more effective.
>
> **Analysis of Representations:** We agree that analyzing attention patterns would be illuminating. In lieu of attention maps, we would point back to Figure 3 itself. The green AIR line shows a much lower (and more stable) cosine similarity between representations of different privilege levels across all layers. This directly demonstrates that AIR is succeeding in helping the model "maintain privilege distinctions" in its intermediate representations, which is the core goal of our method. Additionally, we have added linear probe experiments to test the separability of representations from different layers of the model (see Appendix E). This also shows that AIR provides better separability of intermediate representations encoded with different privilege levels compared to prior works.
>
> **2. Inconsistent performance across training methods**
>
> **Why SFT Can Degrade Utility:** We hypothesize this utility degradation is linked to the fundamental difference between SFT and DPO for adversarial robustness training.
> Supervised Fine-Tuning (SFT) trains the model on a specific target output (the "correct" response) given an adversarial input. This can be a very strong, and potentially brittle, signal. If the model learns too well to reject injected instructions (e.g., by prioritizing the system prompt and ignoring all user data), its performance on benign tasks that require it to use the data segment (the "utility" task in the SEP benchmark) can suffer. This suggests a "robustness-utility trade-off" that SFT struggles to balance.
> Direct Preference Optimization (DPO), in contrast, trains the model on preferences. It learns that the original, correct response $R$ is preferred over the adversarial response $R^{\prime}$. This is a softer, more nuanced signal. It teaches the model what not to do (follow the adversarial instruction) while still reinforcing the correct behavior, allowing it to better preserve general utility.
> Our results consistently show that DPO training (row (a) in Figures 8 and 9) achieves a superior balance, yielding high robustness and high utility.
>
> **Guidance: DPO vs. SFT:** Based on our findings, DPO is the recommended adversarial training method for use with AIR. Across all models and evaluations, the combination of AIR with DPO consistently achieves the best utility-separation tradeoff. While AIR-SFT still provides excellent robustness (often matching or beating other methods in Table 1 and Figure 9b), its potential to degrade utility makes it a less reliable choice.

---

> > ### Author Response · Authors · 2025-11-21
> >
> > **3. Model-specific hyperparameter sensitivity**
> >
> > The performance of AIR is sensitive to the initialization $\sigma$ primarily if the value is too low relative to the magnitude of the intermediate representations ($\vec{x}_{ij}$) it is being added to. Our investigation revealed exactly this: the magnitude of the Qwen model's intermediate activations is approximately 5x larger (geometric mean of L2 norm across all layers) than that of the Llama models (measured across 100 samples from the Alpaca dataset). Therefore, the default $\sigma=0.02$ (effective for Llama) resulted in embedding vectors that were too small to sufficiently modify Qwen's representations, leading to suboptimal robustness. The 5x increase to $\sigma=0.1$ was chosen as the correction to match the activation scale of the target model, which demonstrably improved effectiveness.
> >
> > **Practical guidelines for choosing $\sigma$**
> >
> > While a thorough investigation of the sensitivity of this hyperparameter for different models and layers would be interesting, this requires dozens of training runs. While we lack the resources to conduct this study, we suggest the following practical guidelines to help with the choice of $\sigma$.
> >
> > **Analyze Activation Scale:** Before training, run a few forward passes on a sample of data (e.g., 100 examples from Alpaca) to measure the average magnitude (L2 norm) of the intermediate representations ($\vec{x}_{ij}$) that AIR will augment.
> >
> > **Scale Initialization Accordingly:** Use a baseline model (e.g., Llama-3.1-8B with $\sigma=0.02$) and scale the initialization $\sigma$ for the new model's AIR embedding tables proportionally to its observed activation magnitude.
> >
> > We have added the above guidelines to Appendix B.2
> >
> >
> > **4. Limited evaluation scope**
> >
> >
> > **Single-Turn vs. Multi-Turn:** We agree that multi-turn and agentic settings are critical, as noted in our limitations . Our work focuses on single-turn evaluations consistent with the experimental methodology of prior works [1][2][3]
> >
> > **Static vs. Adaptive Attacks:** We fully agree. We included static attacks (Table 1) as a "sanity check" to show all methods solve these known vulnerabilities . This is precisely why our paper's core evaluation focuses on the much harder, adaptive gradient-based attacks (GCG and Astra). It is on these attacks that the crucial differences between AIR, ISE, and Delimiters become clear, as shown in Figures 7 & 9 and Table 1
> >
> > **Additional static attacks:** We have included evaluations on indirect prompt injections from the BIPIA [4] dataset in the appendix. The results from this dataset provide additional evidence to show that AIR provides higher robustness compared to prior works.
> >
> > **References**
> >
> > [1] Sizhe Chen, Julien Piet, Chawin Sitawarin, and DavidWagner. Struq: Defending against prompt 348 injection with structured queries. arXiv preprint arXiv:2402.06363, 2024.
> >
> > [2] Tong Wu, Shujian Zhang, Kaiqiang Song, Silei Xu, Sanqiang Zhao, Ravi Agrawal, 397 Sathish Reddy Indurthi, Chong Xiang, PrateekMittal, and Wenxuan Zhou. Instructional segment 398 embedding: Improving llm safety with instruction hierarchy. arXiv preprint arXiv:2410.09102, 399 2024.
> >
> > [3] Sizhe Chen, Arman Zharmagambetov, Saeed Mahloujifar, Kamalika Chaudhuri, David Wagner, 350 and Chuan Guo. Secalign: Defending against prompt injection with preference optimization, 351 2025. URL https://arxiv. org/abs/2410.05451, 2024.

---

### Meta-Review · Area_Chair_K1RU · 2026-01-01

**Summary:**

The reviewers raised concerns regarding (1) unexplored scenarios (e.g., multi-turn, retrieval-augmented, and user-as-attacker); (2) the robustness of fine-tuning; (3) the failure of static attacks; and (4) sensitivity to hyperparameter settings.

The authors are encouraged to address these issues to strengthen the work for publication.

**Reviewer Concerns:**

Reviewer PM5M:
- Limited theoretical justification *partially solved*
- Inconsistent performance across training methods *partially solved*
- Model-specific hyperparameter sensitivity *partially solved*
- Limited evaluation scope *partially solved*
---
Reviewer kjgc:
- Validation of motivating hypothesis *partially solved*
- Static attacks fail *appears to be not answered directly*
---
Reviewer Pn8n:
- Utility is measured mostly via AlpacaEval win rates. There is no assessment of factual accuracy or reasoning.
- Although the paper suggests AIR can be applied to direct prompt attacks and agent settings, no experiments verify this. Multi-turn, retrieval-augmented, and user-as-attacker (jailbreak) scenarios remain unexplored.
- There is no visualization of how AIR changes attention patterns across layers.
- The SFT models are fully fine-tuned, while DPO models use LoRA. This mismatch may explain the utility drop seen in Figure 8.
(Reviewer response: The authors have addressed some of my concerns.)

**Reviewer Scores:**

Reviewer PM5M *is likely to keep the score*.
Reviewer kjgc *is likely to keep the score*.
Reviewer Pn8n *replied and will keep the score*.

---

### Decision · Program_Chairs · 2026-01-26

Reject